# Interoperability and Integration: An Updated Approach to Linked Data Publication at the Dutch Land Registry

Alexandra Rowland [1,*], Erwin Folmer [2], Wouter Beek [3] and Rob Wenneker [4]

1   Kadaster & University of Twente, Faculty of Electrical Engineering Mathematics and Computer
    Science (EEMCS), 7500 AE Enschede, The Netherlands
2   Kadaster & University of Twente, Faculty of Behavioural, Management and Social Sciences (BMS),
    7500 AE Enschede, The Netherlands; erwin.folmer@utwente.nl
3   Kadaster & Triply, 1043 BP Amsterdam, The Netherlands; wouter@triply.cc
4   Kadaster, 7311 KZ Apeldoorn, The Netherlands; rob.wenneker@kadaster.nl
*   Correspondence: lexi.rowland@kadaster.nl

**Abstract:** Kadaster, the Dutch National Land Registry and Mapping Agency, has been actively publishing their base registries as linked (open) spatial data for several years. To date, a number of these base registers as well as a number of external datasets have been successfully published as linked data and are publicly available. Increasing demand for linked data products and the availability of new linked data technologies have highlighted the need for a new, innovative approach to linked data publication within the organisation in the interest of reducing the time and costs associated with said publication. The new approach to linked data publication is novel in both its approach to dataset modelling, transformation, and publication architecture. In modelling whole datasets, a clear distinction is made between the Information Model and the Knowledge Model to capture both the organisation-specific requirements and to support external, community standards in the publication process. The publication architecture consists of several steps where instance data are loaded from their source as GML and transformed using an Enhancer and published in the triple store. Both the modelling and publication architecture form part of Kadaster's larger vision for the development of the Kadaster Knowledge Graph through the integration of the various linked datasets.

**Keywords:** linked spatial data; knowledge graph; semantic technologies; interoperability; semantic modelling

## 1. Introduction

The Dutch Cadastre, Land Registry and Mapping Agency, Kadaster (www.kadaster.nl), is the authoritative source of information relating to administrative and spatial data surrounding property and ownership rights in the Netherlands. Kadaster maintains several large key registers of the Dutch government, including the Base Register for Addresses and Buildings (Dutch acronym: BAG), the Base Register for Topography (Dutch acronym: BRT), and the Base Register for Large-scale Topography (Dutch acronym: BGT); all of which are available as open data. In addition, the organisation actively publishes and maintains these and other geospatial assets as linked (open) data and, as part of this effort, and in the spirit of continuous innovation, several of these geospatial assets have now been republished as linked open data following a new approach as discussed in this paper. At the time that this paper was originally submitted for presentation at the 4th International Workshop on Geospatial Linked Data at the EWSC 2021 conference [1], two of these geospatial assets had been published as linked data. As of October 2021, four base registers and a number of external datasets associated with these registers have been published while making use of this new approach: highlighting the efficiency and reproducibility of this approach. The formulation and implementation of this new approach within Kadaster were motivated

by the increased demand for linked data products, both internally and externally, to be delivered in a time- and cost-efficient manner. The availability and ongoing development of linked data technologies and standards have made it possible to introduce an updated and scalable approach to linked data publication to meet this demand. The intention of this paper is to outline the problem context which drove the design and implementation of this approach within Kadaster in more detail and then outline both the novelty in the modelling approach and in the architecture used to publish these datasets. Section 6 of this paper will also discuss the larger vision driving the implementation of this approach within the organisation. Having now made several datasets available through this approach and by having first implementation of the knowledge graph available at Kadaster, this paper will also offer an evaluation and a lessons learned perspective on the approach presented in this paper.

## 2. Problem Context

Dutch governmental geospatial datasets, which are discussed more extensively in the section that follows, are organised as data silos called Key Registers; the integration of which is also generally poor. Users who seek to combine these silos for a given purpose often have to resort to the use of specific tooling which supports this integration or downloading whole datasets and performing the integration of entire datasets. For example, attempting to answer the relatively simple geospatial question 'Which churches were built before 1800 in the city of Amsterdam?' requires the integration of the Key Register for Topography (Dutch acronym: BRT) which includes building types and the Key Register of Addresses and Buildings which includes the building year of each building. To perform this integration, a user either needs to make use of specific GIS tooling (e.g., QGIS or ArcGIS) or download the whole key register and then perform this integration. This problem of easier user integration of base registers is what drove one of the most recent projects carried out by Kadaster, the Integrated User Solution (Dutch: Integrale Gebruiksoplossing (IGO)) project (https://www.geobasisregistraties.nl/basisregistraties/doorontwikkeling-in-samenhang/inspirerend-gebruik). One of the solutions to this integration problem, and one that has been implemented at Kadaster for a number of years, is to provide these data silos as linked open data and perform this integration using associated technologies.

Although several of Kadaster's geospatial assets have been available as linked data for several years and the architecture surrounding this publication has, indeed, been well implemented [2], the network effects of increased uptake in linked data technologies have demanded that an updated, scalable approach to publication be designed and implemented for the publication of these linked datasets. Indeed, there is an increasing demand for linked data services both within Kadaster itself and externally, bringing with it the demand to make linked data available more quickly and in the most (cost) effective manner possible. This demand, coupled with the increasing availability and ongoing development of linked data technologies and standards, has been the driving force behind the innovation of linked data publication within the organisation and has been spurred on by the initiation of projects such as the Integral User Solution. The approach outlined in this paper meets this demand in several ways.

Firstly, this approach, in line with general architectural principles, makes use of existing community libraries (e.g., rdf-validate-shacl (https://github.com/zazuko/rdf-validate-shacl) and (hdt-cpp (https://github.com/rdfhdt/hdt-cpp)), building on top of open-source projects (e.g., Comunica (https://comunica.dev) and ClioPatria (https://cliopatria.swi-prolog.org)) and, therefore, circumventing the need to develop custom, in-house solutions to meet this demand. Similarly, this approach makes use of existing commercial products where they are available in the interest of reducing maintenance costs. Secondly, this approach applies a configuration-over-code principle which ensures that the same pipeline is applied to all linked data publication projects, only configuring components where necessary and it is this aspect of the architecture which has allowed this approach to be replicated on multiple datasets, taking them from source to publication

within a short timeframe. Lastly, the implementation of all relevant components in this design is performed with a streaming approach in mind. In practice, this means that all linked data models are as close to the source model as possible and that the sources selected are able to support streaming functionality in the interest of real-time data delivery. Although this streaming functionality is not yet a working feature in the architecture described, this design inclusion is important within the bigger picture vision for this approach as discussed in Section 6.

In the interest of concretely outlining where the implementation of this approach saw measurable improvements over existing or previous approaches used to publish linked data within Kadaster, it is important to note that the BAG and BGT registries were delivered using this new approach by a small internal team within Kadaster in 9 and 5 weeks, respectively. These are relatively complex linked datasets, each with a complex data model and are large in size. Indeed, each dataset contains between 800 million and 1 billion triples. Where previous approaches could be lengthy in process, this approach highlights improved cost- and resource-effectiveness, strengthening the business case for linked data within an organisation such as Kadaster [3]. The sections that follow outline the concepts and architecture which support this updated approach to linked data publication within Kadaster, including the standards, technologies, and relevant choices made with regards to these during publication of geospatial datasets by Kadaster.

## 3. Native Geospatial Data Sources

There are currently four key registers (complete list: Key Register for Large-Scale Topography (Dutch acronym: BGT), the Key Register for Addresses and Buildings (Dutch acronym: BAG), the Key Register for Topography (Dutch acronym: BRT), and Key Register Kadaster (Dutch acronym: BRK) which are the registration of immovable property rights and the boundaries of national government, provinces, and municipalities in the Netherlands.) maintained by Kadaster and two other/external datasets (complete list: Central Bureau for Statistics (CBS) Key Figures District and Neighbourhood (Dutch acronym: CBS KWB) and the National Service for Cultural Heritage (Dutch acronym: RCE) Monument Register.) that have been transformed and published as linked open data using the approach detailed in this paper. The first of these was the Key Register for Large-Scale Topography (Dutch acronym: BGT), which was transformed and published in November 2020 and has been updated in every quarter of 2021. This asset is a digital map of the Netherlands with included objects such as buildings, roads, bodies of water, and railways. The modelling, updating, and maintenance of this dataset are regulated by Dutch law. The Key Register for Addresses and Buildings (BAG) was transformed and published in February 2021. As the dataset name implies, the dataset includes all buildings and addresses in the Netherlands as well as the attributes associated with these, including house numbers, designations, and main and side addresses. This dataset has a counterpart dataset, namely INSPIRE Addresses (https://www.pdok.nl/introductie/-/article/adressen-inspire-geharmoniseerd-), which is published based on INSPIRE compliance requirements. Among the other datasets published using this approach is the Central Bureau for Statistics in the Netherlands' (Dutch acronym: CBS) Key Figures District and Neighbourhood dataset (Dutch acronym: KWB). This map makes use of the municipal boundaries defined in the Key Register for Land Registry (Dutch acronym: BRK) and provides the aggregated key statistics for neighbourhoods and districts in the Netherlands together with statistics on the proximity of statistics within each area. The fourth Key Register to be transformed to date is the Key Register for Topography (Dutch acronym: BRT). All datasets, including information regarding API availability and querying possibilities, are available in the triple store managed by Kadaster's Data Science Team (https://data.labs.kadaster.nl).

## 4. Knowledge Model vs. Information Model

The first of the new additions to Kadaster's publication of linked data is the explicit distinction between the Knowledge Model and the Information Model, both composing

the larger linked data model for each dataset. This separation reflects the fact that a linked data model must be able to describe the meaning of the data to the outside world using the Knowledge Model, while also describing the organisation-specific aspects of the model in the Information Model. This separation allows the Information Model to be optimised towards the organisation's internal requirements, including specific models and processes relating to an asset, while still allowing the associated Knowledge Model to be optimised towards efficiently supporting external, community standards of publication required for discoverability and interoperability purposes [4]. Since both the internal and external aspects are important for Kadaster's efforts in data publication, this new approach to linked data publication is better able to implement the organisational requirements for linked datasets. Indeed, there are a number of specific reasons in the context of Kadaster that this split was required in designing a new approach to publication.

Firstly, an Information Model for a given asset contains the specific and internal information relating to a given asset. This information includes the properties of the current information systems being used in and in proximity to a given asset, the organisation-specific rules relating to an asset as well as the asset's technical details. For example, the fact that names of municipalities in the Netherlands are not allowed to be longer than 80 characters is not encoded in Dutch law explicitly but is rather a restriction imposed by the internal systems that Kadaster uses to record this information when registered in a key register. Including this information in the Knowledge Model would be incorrect: this maximum length is not part of the semantics for names of municipalities. However, leaving this information out would also be incorrect because the linked data model would not be able to exclude values not accepted by Kadaster's internal registration systems. As such, this information is modelled in the Information Model of a given asset and is represented using Shapes Constraint Language (SHACL) [5] which serves to constrain a given model based on internally defined rules and relationships for a given asset. There are several technical approaches that could be used to formalise such constraints in the Information Model, for example XML Schema or JSON Schema, but SHACL is most optimised for graph-shapes data and linked data. Therefore, SHACL was chosen as the representation language in this project.

Secondly, a Knowledge Model for the same asset defines any generic and interchangeable knowledge that is both important to retain within the organisation but which should also be shared with others. This Knowledge Model also makes it easier to reuse external linked data models with an organisation-specific context. For example, the external SKOS preferred label (skos:prefLabel) property can be used to represent the names of public spaces (Knowledge Model), while also encoding the above-mentioned character limit (Information Model). In Kadaster's implementation of the Knowledge Model, all linked data models make use of RDF(S) [6], OWL [7], and/or SKOS [8] vocabularies and ontologies, amongst others, which serve to make the model relatable to the outside world, invariably in the interest of reusability and interoperability.

As illustrated in Figure 1A,B, the Information and Knowledge Models are not entirely independent of one another and, indeed, are actually mapped to each other when defining and transforming the model. For example, this process maps the SHACL shapes defined in the Information Model to the relevant OWL classes or RDF literals defined in the Knowledge Model. There are two variants to this mapping process, one being the mapping of object properties across the two models (Figure 1A) and the other being the mapping of datatype properties across the models (Figure 1B). Both variants should be completed over the course of a data model transformation into linked data.

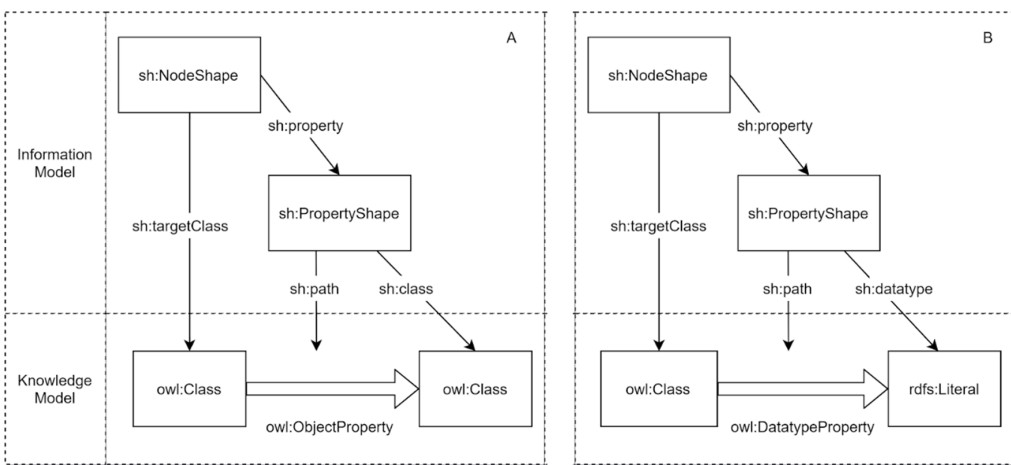

**Figure 1.** Model mapping of object/datatype properties. Schemes follow another format. (**A**) This figure illustrates how the mapping between the Information Model and Knowledge Model is conducted in practice with regards to owl:ObjectProperty; (**B**) this figure illustrates how the mapping between the Information Model and Knowledge Model is conducted in practice with regards to owl:DatatypeProperty.

In the first variant (Figure 1A), the process is almost identical except for the fact that SHACL node and property shapes defined for object properties in a given data model are mapped to the relevant OWL classes and object properties defined in the Knowledge Model. In the second variant (Figure 1B), the datatype properties are defined in the data model by mapping the relevant SHACL node and property shapes for each data type to the relevant OWL class, datatype property, and RDFS literal defined in the Knowledge Model.

In an effort to support better validation of the resultant model, an SHACL validation step has also been applied to the modelling process. This step ensures that the shapes for each object and datatype property in the data model completely validate against instance data. This validation step includes a number of best practices with regards to the modelling of the Information Model centred on the use of closed node shapes [9,10]. This ensures that the model is both as specific as is necessary to ensure that there is a meaningful validation of the model while still allowing correct, but rare, data instances to validate. Table 1 gives examples of how the Information Model is used in addition to the Knowledge Model to implement specific best practices for the data model and for instance data. Notice that the Knowledge Model (column 3) specifies the correct semantics/meaning of the data. The Information Model (column 2) specifies additional structural criteria that are—strictly speaking—not part of the meaning of the data.

**Table 1.** Best practices used for validation of the Knowledge and Information Model.

| Best Practice | Information Model | Knowledge Model | Example |
|---|---|---|---|
| The IRI strategy is correctly implemented. | - Regular expression check on IRI paths. | | - The unique identifier for a building. |
| Linguistic content is readable for human users. | - Correct length (sh:minLength, sh:maxLength).<br>- Valid characters (sh:pattern).<br>- Correct language tags (sh:languageIn). | - Published as human-readable content (e.g., skos:prefLabel).<br>- Semantic characterisation of the intended purpose (e.g., skos:example, skos:definition). | - The human-readable definition of a concept.<br>- The name of a public space. |

**Table 1.** *Cont.*

| Best Practice | Information Model | | Knowledge Model | | Example | |
|---|---|---|---|---|---|---|
| Subsumption relation is correct and easy-to-use. | - | Emit a warning if a class has more than one parent (subsumption relation cannot be shown/navigated as a tree). | - | Semantic characterisation of the subsumption relation (rdfs:subClassOf). | - | A building is a geospatial object. |
| Geospatial object must have geometry. | - | Emit an error if a building has no geometry. | - | Use international OGC standards to semantically characterise the geometry (and its coordinate-reference system). | - | Every building is published with exactly one geometry. |

## 5. Design and Development of Supporting Architectures

The process of converting relational data for a given geospatial asset to linked data is completed in several steps taken during the Extract, Transform, and Load (ETL) process. This process is illustrated in the architecture outline in the figure below (Figure 2) and detailed as follows. In detailing the architecture used in this approach, short discussions around current implementation and areas for improvement will be included intermittently with the view of offering a perspective on the further development of this architecture.

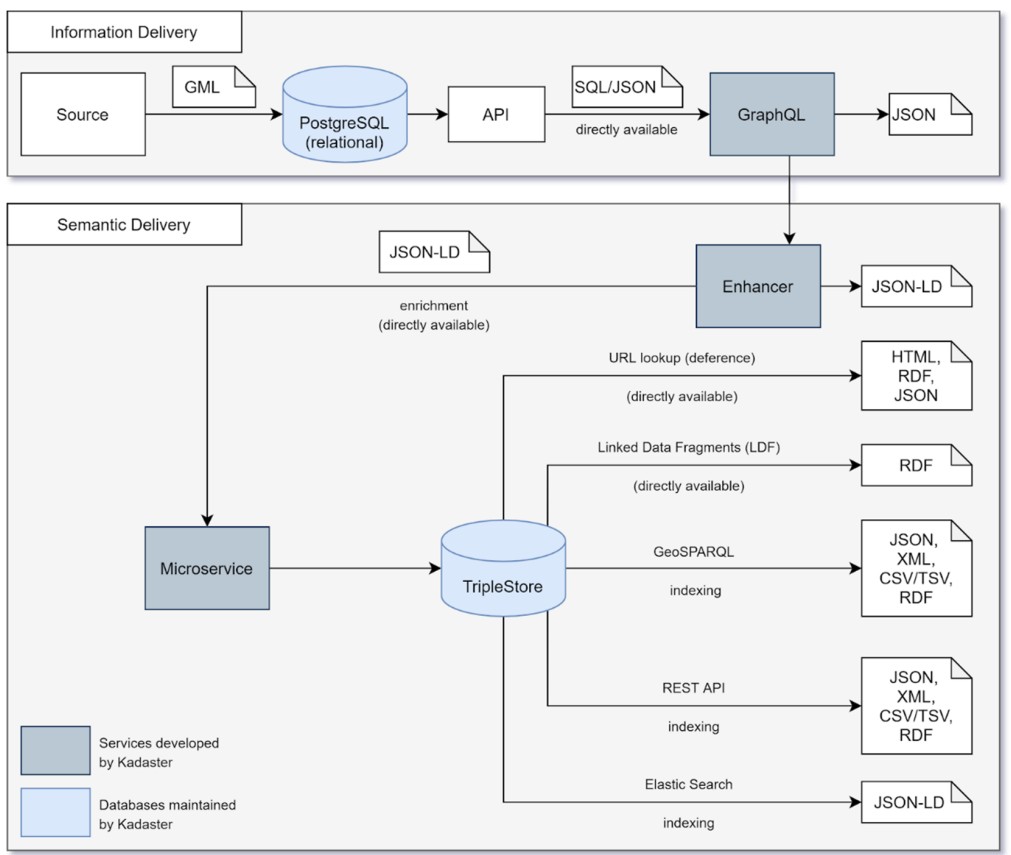

**Figure 2.** Architecture supporting the ETL process which delivers linked data.

The first step loads the relational data from the source, in this case from the Dutch platform for geoservice access (PDOK), to a PostgreSQL database following a Geography Markup Language (GML) indexing step [11]. A GraphQL (https://graphql.org/) endpoint

is then used to access the data delivered through an API from the PostgreSQL database following the delivery and validation of the data model from the end user. In practice, this step is performed by extending the typedefs such that the objects in a data model are correctly described in GraphQL, expanding resolvers to allow objects to be queried with the right parameters and, finally, adding the required SQL queries to the relevant resolvers. Note that this approach is not inherently limited to relational data sources, as a GraphQL endpoint may also be able to deliver from other source types. Additionally, all the GraphQL endpoints that are made available for each data silo are compliant with the Apollo GraphQL standard. Compliance with this standard makes the federation of endpoints easier and, in practice therefore, when different GraphQL endpoints reference each other because key registers contain relationships with each other, for example, Apollo is used to federate these silos.

In the process of retrieving data from the source, data are made available to this pipeline as close to the source as possible to avoid any unnecessary copies of data and to ensure that the topicality of data is as close to real-time data as possible. This 'close to the source' approach ensures that, in modelling the linked data models for each dataset to be converted, no unnecessary or opaque transformation for the data model is made and, as such, the linked data model is completely recognisable to domain experts. In the current implementation of the architecture for each dataset, Kadaster retrieves data for transformation at varying degrees of distance to the source simply based on the current method of delivery implemented by the source itself. Indeed, these methods of delivery range from GitHub publications within a certain timeframe (BAG) to a download of a copy of the data delivered each quarter (BGT) and to the ability to access the data directly from the facility itself (BRT). In the short term, it is likely that the delivery of these sources will be centralised in an information facility through which this pipeline can access the source.

Once the model is available as a GraphQL endpoint, it can be queried by the Enhancer which is an internally developed component which makes it possible to query JSON-LD based on a GraphQL endpoint. To do so, the following steps are required. Firstly, the Enhancer has a set of predefined queries with specific time and/or pagination parameters for each object such that the object is delivered as an endpoint that the microservice can access for the delivery of the JSON-LD [12] results. Configuring the predefined queries used in the Enhancer currently works by adding a GraphQL query under the/query directory in the Enhancer. This query returns instance data based on objects defined in the data model for a particular dataset and can be used in the full load of a dataset such as Linked Data. The specific format of this endpoint is based on an accept header of either application/n-quads or application/ld+json. Note that it is also possible to access the Enhancer endpoint with a POST request containing a GraphQL query. Secondly, a reference to the relevant location of the JSON-LD context for a specific dataset should be defined and this is done for each new dataset that goes through the ETL process. Each key in the JSON-LD context refers to attributes and/or objects in the GraphQL typedefs which support the Enhancer with the conversion of these attributes to a linked data format. As is evident from the previous section, SHACL can be used both for the validation of the data model using example data but also for the validation of the transformed instance data using the data model. Indeed, within this architecture, an SHACL validation step is required to ensure the data delivered from the Enhancer is valid. Finally, in the pipeline of converting a GraphQL delivery to linked data, it is necessary to batch query the total set of objects using predefined queries. These batch queries are contained in a minimal piece of code called the microservice as shown in Figure 2.

The whole ETL process makes use of as little custom code as possible and, therefore, the process does make use of the Apache Airflow (https://airflow.apache.org/) software as a 'handler' which guides the data through the entire ETL process. As illustrated in Figure 2, the microservice fetches data from the enhancer and iterates this process until all data are retrieved as JSON-LD from the original GraphQL endpoint. In practice, this microservice is a task within Airflow in which the Enhancer is addressed serially with limited code and

results are collected as linked data. When all data are validated and loaded into the triple store, which in this case is an instance of TriplyDB (https://triplydb.com), various services can be instantiated, including ElasticSearch, a data browser, a SPARQL [13] endpoint for use in data stories. These can be instantiated within the interface of the triple store itself. In the interest of better accessibility of the linked data models, the data models for each key register are also visualised using the Weaver (https://kadaster.wvr.io/bag2-0/home) tool.

Although the current architecture does not yet include a streaming approach to linked data publication, the architecture itself has been designed in such a way that this approach will become possible with time. The crucial factor in delivering this streaming functionality for linked data at Kadaster is the increased availability of these datasets on the Datahub within the Land Registry. Indeed, because the vision is for this Datahub to be the central source of all cadastral datasets, the Datahub should also comply with a number of principles, including the offering of data in near real time. In order to deliver these principles within its design, a streaming API is likely to be included for each dataset, which enables both pull and subscription-based queries. Connecting the pipeline outlined in this chapter to this streaming API via GraphQL will allow real-time transformation and delivery of linked data for each base registry to the same Datahub environment.

## 6. Vision for Geospatial Data Integration

While advancements in linked data technologies and standards, as well as increased demand for these services, initiated the need for an updated approach to Kadaster's delivery of linked geospatial data, this approach is now also at the centre of Kadaster's ambition to deliver a knowledge graph [14]. In the solution architecture presented in the previous section, the first step towards this knowledge graph is taken in making various key registers available as linked data. These datasets, in striving to keep the linked data model as close to the source data model as possible, are still domain specific. This has an advantage for siloed linked datasets because the data models are recognisable and accessible to domain experts and the governance technique is clear in the management of this data by the dataset owner; factors that are important for the internal governance and ownership of data in various formats within Kadaster.

This 'close-to-the-source' approach to linked data publication is, however, not useful when striving to make data contained within the key registers more accessible to the non-expert user because making use of this data requires some domain knowledge. In practice, non-expert users of such data sources are more interested in information about their homes or living environment and look to interact with the data in terms that are familiar to them based on their surroundings. As such, in delivering the knowledge graph as a result of combining the key registers and other datasets transformed using the solution architecture above, a new, user-friendly data model is used for Kadaster's knowledge graph. The contents of the knowledge graph are the linked datasets for each key register, the digital cadastral map as well as other relevant datasets centred around the theme of a building. The technical process of combining the siloed datasets to form the knowledge graph is illustrated in Figure 3.

As Figure 3 illustrates, Kadaster's knowledge graph is delivered by creating a layer on top of the key registers in their linked data form. The combination of these linked data registries is currently performed by defining a data model using schema.org specifications relating to buildings. The use of schema.org in the first publication of Kadaster's knowledge graph is done for three reasons. Firstly, by architecting the knowledge graph as a layer on top of the siloed linked datasets which make up the graph, the provenance of the original datasets is still available to the end user of the knowledge graph if necessary. Secondly, making use of the schema.org specifications is done in the interest of reusing existing community standards as well as in the interest of supporting external discoverability and interoperability. Additionally, the schema.org specification relating to a building is generally much closer to how non-expert users perceive their environment. Indeed, when looking for information about a house, a user might think in terms of a postcode and

house number and not a number designation. As such, the data model proves to be much more user friendly than complex data models that define the key registers. Access to the current knowledge graph is delivered through REST, GraphQL, GeoSPARQL [15], and ElasticSearch services wherein third-party applications make use of these in delivering geoinformation to the end user [16].

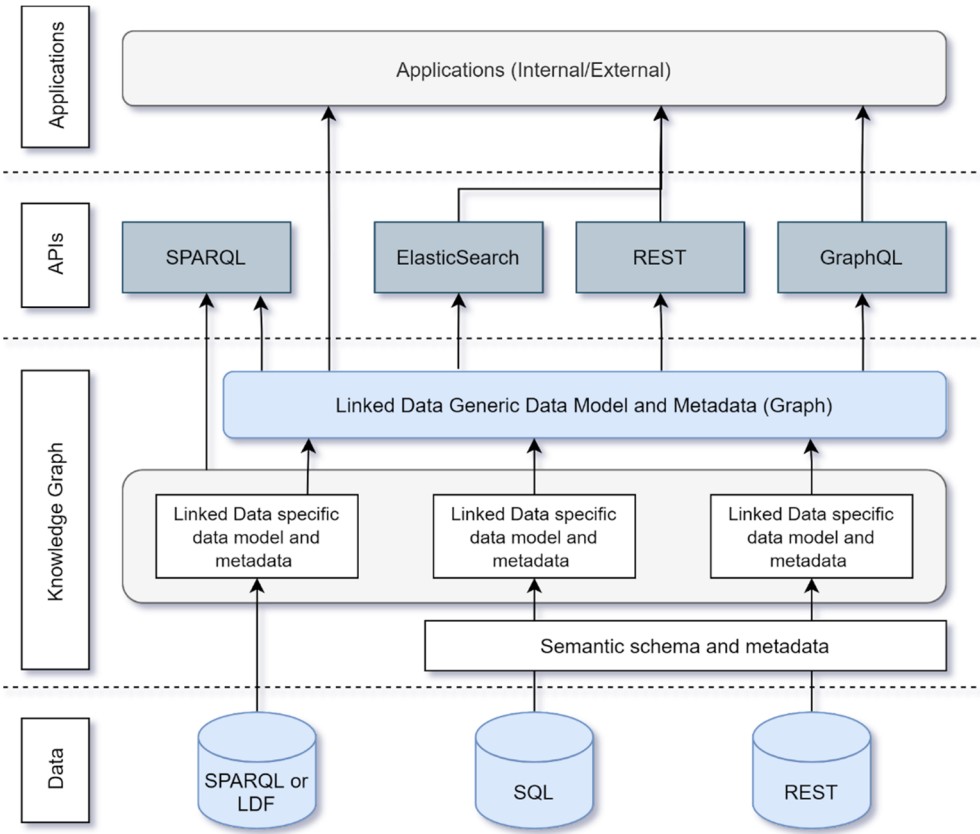

**Figure 3.** Kadaster's vision for the implementation of the knowledge graph.

While the use of the schema.org specification has been useful in developing a proof of concept for the Kadaster knowledge graph, this specification is unlikely to be the final specification used for the data model. This is mostly due to the fact that a future iteration of the knowledge graph is not likely to only contain information about buildings, and the schema.org specification is not (yet) broad enough to consistently model all relevant geoinformation objects (e.g., roads, infrastructure, topographical features). In future iterations of the knowledge graph, Kadaster will work closely with national standards authorities and user groups in defining a data model that is both semantically rich enough to accurately combine and reflect the complexity and richness of information contained within the key registers while also combining the data in such a way that the user-friendly nature of a combined data model is preserved.

## 7. Evaluation and Lessons Learned

In implementing the approach outlined in this paper, various challenges were addressed with regards to the publication of linked data at Kadaster. Indeed, because the initial approach was developed in the context of far fewer linked data libraries and products, it was often necessary to develop the infrastructure required for linked data publication in-house. This accrued a significant amount of legacy infrastructure which was resource intensive to maintain. Over time, this infrastructure also did not reflect the developments being adopted in the wider linked data community with respect to libraries and technologies. As noted throughout the paper, the demand for a more scalable approach as a result

of the network effects related to greater availability of linked data required that the process of linked data publication at Kadaster be innovated. To ensure this scalability and resource efficiency, the new approach turned to existing community standards and commercial technology solutions for the development of the publication timeline in line with general architectural principles which emphasis the need to incorporate existing technologies wherever possible.

As noted in the problem context, the overhaul of the existing approach in favour of the one outlined in this paper has led to the publication and maintenance of several large and complex datasets by a relatively small internal team at Kadaster. This is achieved through a greater emphasis on community and commercial reuse of standard and products which reduce the amount of resources spent on the development and maintenance of the linked data publication infrastructure. Additionally, with the addition of principles such as configuration-over-code as well as dataset publication 'closer to the source' with the streaming approach in mind, the new approach significantly reduces the complexity and resource intensity of previous approaches. Indeed, although the previous approach to linked data publication was successful at the time of its initiation, the development of the related technologies over time drove the need to innovate and, ultimately, the development of the approach developed here.

The successful publication of various key registers and external datasets offers an opportunity to reflect on the approach taken by Kadaster as outlined in this paper. Here, a number of 'lessons learned' can be identified, each of which can be roughly categorised into two themes. The first theme can be summarised as the need for a better fit between the business and technical perspectives on linked data and includes two lessons learned. Firstly, the approach presented in this paper is a technically lightweight, low-cost, and low-code (using existing software solutions) implementation for linked data publication. This has a much better fit with the business perspective and the architecture principles which are generally followed within organisations. For example, the low cost of the implementation complements the fact that there is currently relatively limited usage of these services by end users. Additionally, using existing tools in the publication pipeline is complementary to the reuse of existing software solutions within the organisations based on general 'good practice' architecture principles.

The second lesson to be learned within this theme pertains to the general lack of governance that surrounds existing approaches to linked data publication within organisations. The approach presented in this paper makes a clear distinction between the publication of siloed linked datasets, datasets that are modelled and implemented as close to the source data model as technically possible, and the vision for the implementation of the knowledge graph as a layer on top of these datasets. This distinction from a governance perspective allows the siloed dataset owner to remain responsible for the linked data version of its silo because no significant changes have been made to the data (model) itself and this allows the knowledge graph to fall under a different governance model. This split supports effective governance of linked data within an organisation, both from a technical and business perspective.

The second theme pertains to the issue of data modelling of knowledge graphs within a government context. The first implementation of the knowledge graph at Kadaster was performed by developing a data model based on schema.org. As noted in the previous section, there are various shortcomings of this data model, and this has resulted in the acknowledgement that this integrated data model is necessary and should be developed and maintained by a more authoritative source such as a relevant standards authority or by Kadaster itself. Data modelling in the government context can be somewhat complex due to the various legal and governance requirements that surround a data model. Developing a new, integrated data model for the knowledge graph can take time and the question of who should be responsible for the development, delivery, and maintenance of the model remains an open question. As such, the biggest hurdle for the delivery of an integrated data model in the government context may be the complexity surrounding data modelling

activities; a hurdle which has the potential to delay the widespread application of these graphs for coming years.

## 8. Conclusions

Kadaster, the Dutch National Land Registry, has recently implemented an updated approach to linked data publication of their geospatial assets in response to growing demand for linked data services and the pressing need to innovate existing approaches to meet scalability requirements. Building on existing experience with the publication of their base registries as linked data, Kadaster has made use of existing community technologies and standards as well as available commercial products to define an approach which delivers linked data assets in a timely, cost-efficient manner and with increased reusability across projects. This approach forms part of a larger vision to deliver a knowledge graph centred around the 'Building' theme where both this larger vision and the principles applied to this central approach to transformation of the base registries are implemented in the interest of better geospatial data integration, interoperability, and discovery. Although innovative, Kadaster's effort to improve geospatial findability and linkability has not been conducted in isolation and highlights a general need for better spatial interoperability on a national level [17,18] and between (European) countries [19] and reusability of this data in various contexts [20].

**Author Contributions:** Conceptualization, Alexandra Rowland and Wouter Beek; Methodology, Alexandra Rowland, Erwin Folmer and Wouter Beek; Supervision, Erwin Folmer; Writing—original draft, Alexandra Rowland; Writing—review & editing, Erwin Folmer, Wouter Beek and Rob Wenneker. All authors have read and agreed to the published version of the manuscript.

**Funding:** This research received no external funding.

**Conflicts of Interest:** The authors declare no conflict of interest.

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
