# Peer review of "Interoperability and Integration: An Updated Approach to Linked Data Publication at the Dutch Land Registry"

_ijgi, doi:10.3390/ijgi11010051_

Round 1
Reviewer 1 Report
The paper presents an approach to dataset modeling and implementation architecture and forms part of Kadaster’s larger vision for the development of its knowledge graph.
The paper is quite well organized and it follows a compelling narrative, moving section by section through the proposed Kadaster.
Some general remarks:
- The abstract should be rewritten, it does not fully reflect the content of the paper.
- There is no clear motivation in Section 1.
- Terms from Semantic Web and Linked Data appear throughout the paper. It is worth adding a section or a paragraph showing preliminaries.
- This point is related to point 3, there are many abbreviations of different standards, RDF, JSON-LD, OWL, SHACL, SPARQL, GeoSPARQL, JSON Schema, XML Schema, etc. Please cite them. The same applies to protocols, formats, query languages, and patterns, e.g. REST, GraphQL
- Some footnote parts should be included in the references, there is scientific literature for some of them.
- It seems to me that the References are made incorrectly, they do not follow the journal's guidelines, e.g., the fourth item, as the name suggests, refers to the W3C and the link takes you elsewhere.
- There is no evaluation in the paper.
Author Response
Dear reviewer,
Thank you for your valuable feedback. Please see below my response to your comments.
- This has been rewritten to reflect the paper more closely.
- Added.
- We have not added an entire section because we presume that the reader audience for this paper will likely have some familiarity with the concept of the semantic web.
- We have, however, added the requested citations for all standards and best practices mentioned in the paper.
- All citations referring to standards and best practices have been moved to the references and software, programming languages and/or libraries have remained as footnotes. We have assumed that XML and JSON formats are not a part of general knowledge.
- This has been adjusted.
- This has been added as part of an extended 'Evaluation and Lessons Learned' section.
Kind regards
Reviewer 2 Report
The presented paper is supposed to describe a new approach to dataset modelling and implementation architecture as part of national land registry - Kadaster.
First of all, it is hard to classify whether the presented paper is a research paper or a deployment report. I do not see a clearly stated research problem; there is only an integration problem.
The authors also mention several time the new approach do data publishing within their registries. Unfortunately, there is no detailed description of both and the old one is particularly missing. This means that the assessment of the progress is hard. For the new approach we learned that it is based on the open source software (reuse), applied configuration-over-code principle, and is done with streaming approach in mind. I was particularly interested in this streaming approach but no details were provided.
The processing itself was not described in greater detail either. In figure 2 what is the most interesting in the architecture is only hidden behind a single box “Enhancer”. I would love to learn more what kind of transformations are conducted there, what additional sources are used. Even the way how integration shall be conducted is unclear. Although we know registries, we do not get any details on data models (or information model as the authors prefer to call it).
One of the new stuff in the method seems to be SHACL. Unfortunately, it is not presented appropriately, and it is best reflected in figure 1. There is no significant difference between the figures and the readers may not feel what SHACL is all about. Moreover, the authors tried to delineate the difference between information model and knowledge model in one figure, what was also not successful.
In many places the authors emphasize that the new approach makes data available to the processing pipeline or keep linked data model “as close to the source as possible”. This is also something not clearly defined. What does it mean close (what metric)? No transformations? No latency? No intermediaries? The similarity of models? Or maybe explicit constraints defined in SHACL?
The paper is concluded with lessons learned and they are basically OK. I would just expect more technical content within the middle part of the paper.
Author Response
Dear reviewer,
Thank you for your valuable feedback. We have addressed your comments in the following ways:
- Indeed, the paper presents less a research paper and more as a development report. To address this, we included the problem context section which outlines that the new approach stemmed from a need to upgrade the existing approaches to meet the time, cost and urgency demands of linked data publication. We have strengthened the motivation in the introduction and extended the problem context slightly so we hope this problem context is now clearer. The problem at hand is not, however, an integration problem but more a problem of better/more efficient linked data publication. When this is done well, which we hope to have highlighted, integration also becomes easier.
- We acknowledge that the evaluation of this new approach was indeed difficult in the first submission. To rectify this, we have extended the lessons learned chapter to include an evaluation of the approach based on what has gone before. We have also extended the streaming section to provide more details as requested.
- We have added more details on the enhancer and the integration is addressed in more detail in the second to last chapter. A discussion on the specifics of the data model are considered out of scope for this paper beyond a short discussion in chapter 3 but we have provided links to these models as footnotes.
- Clarifications on 'close to the source' have now been provided.
Kind regards.
Reviewer 3 Report
This is an interesting paper with a novel approach to linked data publication at land registries. I do recommend this paper for publication.
Author Response
Dear reviewer,
Thank you for your feedback. Your review is appreciated.
Reviewer 4 Report
It is nicely written paper about the implementation of an linked open data strategy for the geospatial data in the netherlands. It shows how the publication process has been set up and how the information and knowledge models were designed to account for internal orgnisational and external data users needs.
I found only one very little minor error: The last headline should be numbered 8, there is two 7 in the paper. So it is more editorial.
Author Response
Dear reviewer,
Thank you for your feedback. I have made the small change identified.
Round 2
Reviewer 1 Report
The authors have successfully addressed all the issues pointed in the first round of the review. I am happy to recommend an acceptance.